# Smoke and Mirrors in Causal Downstream Tasks

**Riccardo Cadei**[1], **Lukas Lindorfer**[1], **Sylvia Cremer**[1], **Cordelia Schmid**[2], and **Francesco Locatello**[1]

[1]Institute of Science and Technology Austria
[2]Inria, Ecole normale supérieure, CNRS, PSL Research University

## Abstract

Machine Learning and AI have the potential to transform data-driven scientific discovery, enabling accurate predictions for several scientific phenomena. As many scientific questions are inherently causal, this paper looks at the causal inference task of *treatment effect estimation*, where the outcome of interest is recorded in high-dimensional observations in a Randomized Controlled Trial (RCT). Despite being the simplest possible causal setting and a perfect fit for deep learning, we theoretically find that many common choices in the literature may lead to biased estimates. To test the practical impact of these considerations, we recorded ISTAnt, the first real-world benchmark for causal inference downstream tasks on high-dimensional observations as an RCT studying how garden ants (*Lasius neglectus*) respond to microparticles applied onto their colony members by hygienic grooming. Comparing 6 480 models fine-tuned from state-of-the-art visual backbones, we find that the sampling and modeling choices significantly affect the accuracy of the causal estimate, and that classification accuracy is not a proxy thereof. We further validated the analysis, repeating it on a synthetically generated visual data set controlling the causal model. Our results suggest that future benchmarks should carefully consider real downstream scientific questions, especially causal ones. Further, we highlight guidelines for representation learning methods to help answer causal questions in the sciences.

Code: `https://github.com/CausalLearningAI/ISTAnt`
Data: `https://doi.org/10.6084/m9.figshare.26484934.v2`

## 1 Introduction

Uncovering the answer to many scientific questions requires analyzing massive amounts of data that humans simply cannot process on their own. For this reason, leveraging machine learning and AI to help answer scientific questions is one of the most promising frontiers for AI research. As a result, AI is now predicting how proteins fold [Jumper et al., 2021], new materials [Merchant et al., 2023], precipitation forecasts [Espeholt et al., 2022], and animal behaviors [Sun et al., 2023]. Even predicting counterfactual outcomes for treatment effect estimation appears to be possible [Feuerriegel et al., 2024]. In scientific applications, these predictions are often incorporated into broader analyses to draw new physical insights. In this paper, we focus on the problem of estimating the strength of the causal effect of some variable on another, which is a common type of question across disciplines [Robins et al., 2000, Samet et al., 2000, Van Nes et al., 2015, Runge, 2023].

While our discussion and conclusions are general, we follow a simple real-world example throughout the paper: behavioral ecologists want to study the social hygienic behavior in ants and, thereby, the ability of the insects to remove small particles from the body surface of exposed colony members. Such grooming behavior performed by nestmates plays an important role in restoring a clean body surface of the contaminated individual, which, in case of infectious particles being groomed off, assures the health of the individual and prevents disease spread through the colony's [Rosengaus et al., 1998, Hughes et al., 2002, Konrad et al., 2012]. To study whether different microparticles differ systematically in their induction of grooming behavior, the biologists thus perform an experiment under

controlled conditions, where a focal worker ant is treated randomly with either of two microparticle types, and the behavior of two untreated colony members towards the treated ant is filmed in multiple replicates. This is followed by detailed behavioral observation to quantify ant activity, as well as statistical data analysis to determine if treatment has a significant effect. This step could obviously be entirely replaced with deep learning, dramatically accelerating the workflow. In fact, many data sets and benchmarks have been proposed with the specific reason of supporting downstream science in behavioral ecology and biology [Sun et al., 2023, Beery et al., 2018, Kay et al., 2022, Chen et al., 2023] and other scientific disciplines [Beery et al., 2022, Lin et al., 2023, Moen et al., 2019].

Our paper questions the simplicity of this narrative in both theory and practice. While we take experimental behavioral ecology as an example for our motivation and experiments, our theoretical results and experimental conclusions are general, and we expect them to be applicable across disciplines. Our key contributions can be summarized as follows:

- We theoretically show how many design choices can affect the answer to a causal question, from the data used for training, the architecture choices, and even seemingly innocuous standard practices like thresholding the predictions into hard labels, or using held out accuracy for model selection (a common practice in many AI for science benchmarks, e.g., [Sun et al., 2023]). To facilitate future research on representation learning for causal downstream tasks, we *formulate the representation desiderata to obtain accurate estimates for downstream causal queries* together with best practices.

- To showcase the practical impact of these design choices, we design and collect a new dataset, *ISTAnt*, from a real randomized controlled trial, reflecting a real-world pipeline in experimental behavioral ecology, which we will release to accelerate research on representation learning for causal downstream tasks. To the best of our knowledge, this is the *first real-world data set specifically designed for causal inference downstream tasks from high-dimensional observations*.

- On our dataset, we *fine-tune 6 480 state-of-the-art methods* [Dosovitskiy et al., 2020, Zhai et al., 2023, Radford et al., 2021, He et al., 2022, Oquab et al., 2023] in the few- and many-shot settings. Empirically, we confirm that the seemingly innocuous design choices like which samples to annotate, which model to use, whether or not to threshold the labels, and how to do model selection have a major impact on the accuracy of the causal estimate. Since our ground-truth estimate of the causal effect depends on the trial's design, we propose a *new synthetic benchmark* based on MNIST [LeCun, 1998] controlling for the causal model, and we replicated the analysis.

## 2 Setting

We consider the RCT setting, where binary treatments $T$ are randomly assigned within an experiment with controlled settings $\boldsymbol{W}$. In many applications, the outcome of interest $Y$ is not measured directly. Instead, it is collected in high-dimensional observations $\boldsymbol{X}$ – e.g., frames from a video of the experiment. Our goal is to estimate the causal effect of $T$ on $Y$, which is quantified by the estimation of the Average Treatment Effect (ATE), i.e.:

$$ATE := \mathbb{E}[Y|do(T = 1)] - \mathbb{E}[Y|do(T = 0)]. \tag{1}$$

Assuming an RCT (i.e., Ignorability Assumption [Rubin, 1978]) is the ideal setting for causal inference because the ATE directly identifies in the Associational Difference (AD), i.e.,

$$AD := \mathbb{E}[Y|T = 1] - \mathbb{E}[Y|T = 0]. \tag{2}$$

However, annotating $Y$ from the high-dimensional recordings $\boldsymbol{X}$ requires costly manual annotations from domain experts. Leveraging state-of-the-art deep learning models, we can hope to alleviate this effort. Instead of labeling all the data, we only partially annotate it. We introduce a binary variable $S$, indexing whether a frame is annotated by a human observer or not. We denote the annotated samples with $\mathcal{D}_s = \{(\boldsymbol{W}_i, T_i, \boldsymbol{X}_i, Y_i) : S_i = 1\}_{i=1}^{n_s}$ and the not annotated ones with $\mathcal{D}_u = \{(\boldsymbol{W}_i, T_i, \boldsymbol{X}_i) : S_i = 0\}_{i=1}^{n_u}$. We use $\mathcal{D}_s$ to train or fine-tune a deep learning model to estimate the labels on $\mathcal{D}_u$. Next, we leverage the Ignorability Assumption on the full RCT to identify the ATE in the AD and consistently estimate it. Ideally, it would be most useful if $\mathcal{D}_s = \emptyset$, but for the purpose of this paper, we assume that at least some samples are annotated, for example, during quality controls.

Besides the clear statistical power considerations, recovering the full RCT enables the identification of the causal estimands. Estimating the ATE only on $\mathcal{D}_s$ may not be feasible even if one aims to

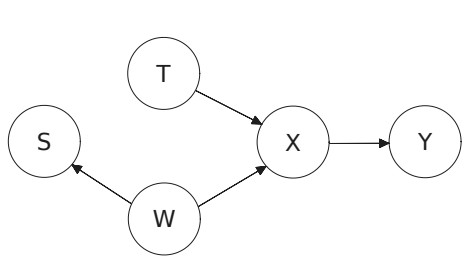

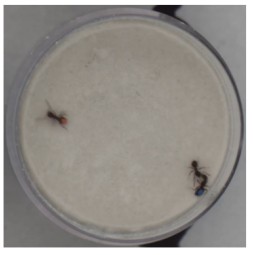

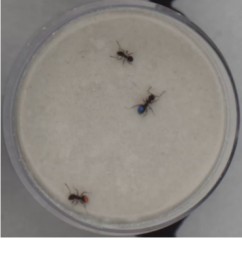

(a) Grooming (blue to focal)   (b) No Action

Figure 1: Causal Model for generic partially annotated scientific experiment: $T$ treatment, $W$ experimental settings, $X$ high-dimensional observation, $Y$ outcome, $S$ annotation flag.

Figure 2: Examples of high-dimensional observations $X$ with corresponding annotated social behaviour $Y$ from ISTAnt (ours).

adjust for $W$ due to possible violations of the Positivity Assumption (i.e., $0 < P(T = 1|W = w) < 1 \quad \forall w : \mathbb{P}(W = w) > 0$). In principle, $S$ should be assigned randomly (independent from any other variable), but for practical reasons, it is often a function of the experiment settings $W$. For example, when annotating grooming in behavioral experiments, experts annotate experiment by experiment, marking the beginning and end of each behavior event, allowing for some selection bias. The experimental setup can be described with the causal model in Figure 1, where we omit the corresponding exogenous random noises for simplicity. For simplicity of exposition, we will now assume a binary outcome, but all the following results naturally generalize to the continuous case.

**Motivating Application and the ISTAnt Dataset.**  Ants show strong hygiene behaviors and remove any particles that attach to their body surface, including dust, dirt, and infectious particles. In a process termed "grooming", they use their mouthparts to pluck off adhering particles, collect and compact them in a pouch in their mouth, and later expulse them as pellets. As a social behavior, ants groom one another to keep all colony members clean and healthy. To understand how social insects like ants may react to changes in their ecosystem, it is of great interest to research in collective hygiene how different particles differentially affect the intensity of grooming by colony members. For this purpose, we recorded groups of three *Lasius neglectus* worker ants interacting in a controlled environment, where we treated one *focal ant* by applying either of two microparticle types to its body surface and observed the grooming activity of the other two towards it. Our exemplary task is to estimate the causal effect of the microparticle type on ant behavior. Sample frames of these recordings used to build our new benchmark are reported in Figure 2.

**Key research question.**  Predicting animal behavior is a standard machine learning and computer vision task [Sun et al., 2023, Chen et al., 2023]. At the same time, we hope to use these predictions within the context of a causal downstream task. In this paper, we question whether the naive application of deep learning methods leads to consistent estimates that can be used to draw scientific insights, even if the data we collect is ideal, i.e., a randomized controlled trial. Likewise, in causal inference, the factual effects are always assumed to be given, and the statistical consideration of using machine learning to estimate them is missing.

## 3 Biases in downstream ATE estimation from ML pipelines

In this section, we formalize a model's bias for a downstream Treatment Effect Estimation and its relationship with (vanilla) prediction accuracy measures. We then highlight possible sources of biases from both the data and the model.

**Definition 3.1** (Treatment Effect Bias). *Let $f : \mathcal{X} \to \mathcal{Y}$ a model for $\mathbb{E}_Y[Y|X = x]$. We define the* **treatment effect bias** *of $f$ w.r.t. a treatment $T$ on an outcome $Y$ and a signal $X$ as:*

$$TEB := \left( \underbrace{\mathbb{E}_{X|do(T=1)}[f(X)] - \mathbb{E}_{Y|do(T=1)}[Y]}_{\textit{Interventional Bias under Treatment}} \right) - \left( \underbrace{\mathbb{E}_{X|do(T=0)}[f(X)] - \mathbb{E}_{Y|do(T=0)}[Y]}_{\textit{Interventional Bias under Control}} \right) \quad (3)$$

*f is treatment effect unbiased if $TEB = 0$, i.e., the difference among the systematic errors per intervention (over/under estimating) compensates, or directly, the ATE on the predicted outcomes equals the true ATE (despite possible misclassification).*

> **Lemma 3.1** (Informal). *Assuming the setting described in Section 2. A predictive model $f$ for the factual outcomes with accuracy $1\text{-}\epsilon$ can lead to $|TEB(f)| = \frac{\epsilon}{\min_t P(T=t)} \geq 2\epsilon$, which invalidates any causal conclusion when the ATE is comparable with $\epsilon$ and/or the dataset is unbalanced in T.*

A formal statement and proof for Lemma 3.1 is reported in Appendix A.1. Lemma 3.1 explicits that misclassification can lead to biased causal conclusion, but not necessarily. Clearly, if the prediction accuracy is perfect (i.e., $\epsilon = 0$), also the objective of treatment effect estimation is perfect. However, for each error rate $\epsilon > 0$, several predictions with different treatment effect biases are possible, from 0 to the worst-case scenario $\frac{\epsilon}{\min_t P(T=t)}$, which drastically invalidates any causal conclusion for $\epsilon \gg 0$ or strongly unbalanced dataset with respect to the treatment assignment. Accuracy and similar metrics do not provide a full picture of the goodness of a model for such a downstream task.

Due to the Fundamental Problem in Casual Inference [Holland, 1986], we cannot estimate the treatment effect bias directly. By design (i.e., Ignorability Assumption), the interventional expectations are identified in the conditional ones on the whole population, but not on $\mathcal{D}_s$ individually due to the effect modifications activated by conditioning on $S$. Still, in practice, a validation set, ideally Out-of-Distribution from the training sample in $\mathcal{D}_s$, can be considered to approximate the TEB.

**Links to Fairness**  This idea of enforcing similar performances (or at least similar systematic errors) among the treated and controlled groups can be revisited in terms of fairness requirements [Verma and Rubin, 2018]. In particular, it strictly relates to Treatment Equality [Berk et al., 2021], where the ratio of false negatives and false positives for both treated and control groups is enforced to be the same, while in TEB we measure the difference, but in a similar spirit. In our setting, the difference is actually a more stable measure since the ratio can be ill-defined when the number of false positive predictions approaches 0. This discussion leaves open where the bias originates and, in the fairness literature, this is reflected in the distinction between bias preserving and bias transforming metrics [Wachter et al., 2021]. For our purposes, the data as a whole is assumed unbiased in principle since we assume an RCT, but the sampling scheme $S$ could introduce bias in the training data. Orthogonally, the model choices can amplify existing data biases differently or even introduce new ones.

**Data bias from sampling choice**  From the assumed causal model illustrated in Figure 1, we have that $\mathcal{P}^{(X,Y)|S=0}$ generally differs from $\mathcal{P}^{(X,Y)|S=1}$. Indeed, conditioning on $S$ acts as an effect modification on $X$ and $Y$. It follows that the risk in predicting $Y$ over the annotated population can differ from the expected risk over the whole population, i.e.:

$$\mathbb{E}_{(\boldsymbol{X},Y)|S=1}[\mathcal{L}(f(\boldsymbol{X}),Y)] \neq \mathbb{E}_{(\boldsymbol{X},Y)}[\mathcal{L}(f(\boldsymbol{X}),Y)]. \tag{4}$$

Due to this distribution shift, we should expect some generalization errors at test time through empirical risk minimization even if $n_s \to \infty$. It follows that the Conditional Average Treatment Effect (CATE) estimate for the experimental settings poorly represented in $\mathcal{D}_s$ can introduce bias in $\mathcal{D}_u$.

> *Mitigation: Randomly assigning $S$ is crucial to suppress any backdoor path and avoid generalization errors. Model selection should also take into account the TEB. Although we cannot estimate it directly, a validation set, ideally Out-of-Distribution in $\boldsymbol{W}$, should be considered to bound the TEB, replacing the interventional distributions with the corresponding conditionals.*

**Model bias from the encoder choice**  Since $\boldsymbol{X}$ is high-dimensional, we decompose the model $f$ in $h \circ e$, where $e$ is an encoder potentially pre-trained on a much larger corpus through a representation learning algorithm and $h$ is a simple decoder (e.g., multi-layer perceptron) for classification. A good representation should be both sufficient and minimal [Achille and Soatto, 2018]. If a representation is only sufficient, redundant information from $\boldsymbol{W}$ or $S$ could be preserved, potentially leading to systematic errors on $\mathcal{D}_u$ due to spurious correlations and the abovementioned covariates shift. Frozen state-of-the-art models are most likely not minimal for our task, making the sampling choices even more relevant. If the representation is not sufficient, then it is biased by definition.

*Mitigation: Before deploying a new backbone, one should attempt to quantify its biases. If needed, new methodologies to mitigate this bias during adaptation should be investigated. Overall, models with lower bias may be preferable even if they have lower accuracy.*

**Discretization Bias**  We can encounter a final source of bias in post-processing the predictions. Indeed, despite the majority of the classification methods directly modeling the conditional expectation $\mathbb{E}[Y|\boldsymbol{X} = \boldsymbol{x}]$, we could naively be tempted to binarize this estimate to the most probable prediction or setting a fixed threshold. See indeed how the default choices for the `predict` module, even in established libraries, e.g., Logistic Regression implementation in `sklearn.linear_model` [Pedregosa et al., 2011], is to output the most probable prediction directly. Similarly, even `econML`, the most popular library for causal machine learning [Battocchi et al., 2019], allows for binary outcome prediction methods. Despite being apparently innocent and common practice in classification, discretizing the conditional expectation is biased for downstream treatment effect estimation.

> **Theorem 3.1.** *[Informal] Let a binary classification model converge to the true probability of the outcome given its (high-dimensional) signal. Then, its discretization (i.e., rounding the prediction to {0,1} with a fixed threshold) also converges, but to a different quantity with a different expectation. It follows that, for causal downstream tasks from ML pipelines, discretizing the predictions biases the ATE estimation.*

A formal statement and proof of Theorem 3.1 is reported in Appendix A.2. It shows that even if we rely on a consistent estimator of the factual outcome for each subgroup, its discretization would still converge but on a different quantity, i.e., it is biased. There is then no reason to discretize a model for $\mathbb{E}_{\boldsymbol{X}}[Y|\boldsymbol{X}]$ if we can model it directly, e.g., using sigmoid or softmax activation [Senn and Julious, 2009, Fedorov et al., 2009]. Likewise, if there is uncertainty over human annotations (e.g., because multiple raters disagree), the soft label should be used and not the majority one.

**Example 1.** *To intuitively visualize this result, consider a generative process following the causal model introduced in Figure 1. Let $\hat{f}$ a model for $\mathbb{E}[Y|X = x]$ trained by logistic regression over $n$ samples and $\hat{f}^*$ its discretization. Let the Empirical Associational Difference (EAD) of $\hat{f}$ converge to its AD, then the EAD of $\hat{f}^*$ still converges but to its own AD, which significantly differs (depending on the randomness in $\mathbb{P}(Y|X)$ mechanism). In Figure 3, we report the results of a Monte Carlo simulation for an instance of this generative process. A full description of the Structural Causal Model and theoretical derivation of the limits is reported in Appendix B.1.*

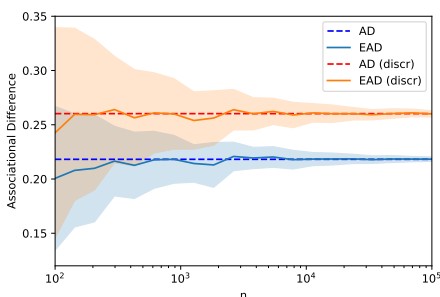

Figure 3: Monte-Carlo simulation of the discretization bias' convergence result.

*Mitigation: Never arbitrarily discretize predictions for downstream treatment effect estimation.*

## 4  Related Works

**Representation learning for scientific applications**  The setting we consider is very common, and there are several benchmarks that have studied representation learning as a means to help domain experts in sciences, for example, [Sun et al., 2023, Beery et al., 2018, Kay et al., 2022, Chen et al., 2023] in ecology alone. However, these works focus on downstream prediction accuracy following standard machine learning evaluation practices, which do not necessarily indicate good downstream causal predictions. One notable positive example is Beery et al. [2018], as computing the prediction accuracy separately for different locations allows us to estimate the bias of the model. Overall, we argue that when the ultimate purpose of training a machine learning model is to support scientists in answering some research question that is causal in nature, the specific question should be part of the design and evaluation of the benchmark. For this reason, our paper uniquely starts from the causal downstream task. Only then can we formalize the properties that methods should have in order to do well and be

useful in answering the overarching scientific question. To the best of our knowledge, ours is the first real-world computer vision data set with an associated well-defined and real causal downstream task.

**Causal representation learning**  In our analysis (both theoretically and experimentally), we focused on traditional representation learning algorithms, but there is a whole community interested in identifying causal variables from high-dimensional observations [Schölkopf et al., 2021]. Superficially, identifying $Y$ may be useful to estimate the ATE. However, all existing methods seem to cover two main classes of assumptions that are unfortunately inapplicable to our setting. Interventional methods [Ahuja et al., 2023, Buchholz et al., 2023, Squires et al., 2023, Varici et al., 2023, Zhang et al., 2024] require intervening on the behavior $Y$, which is practically impossible and, even if we could, then we would not need to identify it. Multi-view approaches [Ahuja et al., 2022, Brehmer et al., 2022, Locatello et al., 2020, von Kügelgen et al., 2021, Yao et al., 2024] would require access to positive and negative pairs of samples with respect to $Y$. However, it is not clear how to construct such pairs in our setting without knowing $Y$ already. Further, all these approaches only cover continuous variables. A notable exception is Kivva et al. [2021], which covers discrete variables but has a non-degeneracy assumption (Assumption 2.4) that is severely violated in our case (i.e., most pixels are not affected by the behavior variable because ants are small). For these reasons, despite having a very clear causal downstream task, we had to resort to classical representation learning algorithms that are not identifiable. We hope that our data set can serve as a new real-world benchmark for developing algorithms with realistic assumptions that can be applied in practice.

**Other Related Works**  In causal inference, only Chakrabortty et al. [2022] shows how to use semi-supervised learning to perform imputation on missing effect annotations. Unfortunately, their setting is comparatively very low-dimensional (observations are  200 binary variables). Instead, we consider high-dimensional real-world images in a representation learning setting, which introduces additional new challenges as described in Section 3. Remarkably, they do not discute discretization bias. Curth et al. [2024] already mentioned that the Positivity and Ignorability/Unconfoundness Assumptions are critical for using machine learning in the context of ATE estimation. However, their work does not explain precisely how confounding effects can arise in the representation learning setting, which we thoroughly addressed. Close in spirit to our discussion are [Angelopoulos et al., 2023, Zrnic and Candès, 2024], considering the role of predictions in statistical estimates. Our setting is related but additionally motivated by the hope of leveraging *causal* identification properties on the prediction-powered dataset.

## 5   Experimental setup

We validate the theoretical results from Section 3 on our new real-world dataset. We assume $\mathcal{D}_s \cup \mathcal{D}_u$ being a full RCT, and we compare the treatment effect biases among several design choices in annotating and modeling. Overall, we fine-tuned 6 480 different models and tested all the mitigations proposed. We then replicate the experiments on CausalMNIST, a new synthetic benchmark we propose that allows controlling for the causal effect.

### 5.1   New real-world dataset: ISTAnt

We applied microparticles to the body surface of a (focal) *Lasius neglectus* worker ant and recorded the behavioral reaction this treatment elicits in two other worker ants from the same colony. To distinguish between the treated individual and the untreated two nestmates, the latter had been color-coded by a dot of blue or orange paint, respectively, before the experiment. We used two different microparticle treatments to compare grooming responses by the nestmates between treatment types, assigning them at random (i.e., RCT). For five batches, we simultaneously filmed nine ant groups of three ants each under a single camera setup in a custom-made box with controlled lighting and ventilation. In total, we collected 44 videos[1] of 10 minutes at 30fps each for a total of 792 000 frames annotated following a blind procedure, and we run the analysis at 2fps for a total of 52 800 frames. More details about the experiment design are reported in Appendix C. We remark that this is the first real-world data set for treatment effect estimation from high-dimensional observations, which we will release to accelerate future research. Since it encompasses a real-world scientific question, we can, at

---

[1]One video was discarded for analysis since a leg of one of the two nestmates got stuck in the dot of the color code, impairing its behavior.

best, enforce the Ignorability Assumption by design in the trial. We do not have actual control over the underlying causal model and the causal effect. We take the treatment effect estimation computed with the expert annotations as ground truth.

**Annotation Sampling**  Annotating frames individually is significantly more expensive in terms of time and not adopted in practice. The practical gold standard through current software for human annotation is per-video random annotation, where only a few videos taken at random are fully annotated. We compared this criterion with per-video batch ($W_1$) and per-video position ($W_2$) annotation criteria, where only the videos in certain batches or positions were considered in $\mathcal{D}_s$. For each of the three criteria, we further considered a many-shots ($\mathcal{D}_s \gg \mathcal{D}_u$) and a few-shots setting ($\mathcal{D}_s \ll \mathcal{D}_u$). Details about the dataset splitting per annotation criteria are in Appendix D.1.

**Modeling**  We modeled $f$ as a composition of a freezed pre-trained encoder $e$ and a multi-layers perceptron $h$ fine-tuned on $\mathcal{D}_s$. For the encoder, we compared six different established Vision Transformers (ViT), mainly varying the training procedure: ViT-B [Dosovitskiy et al., 2020], ViT-L [Zhai et al., 2023], CLIP-ViT-B,-L [Radford et al., 2021], MAE [He et al., 2022], DINOv2 [Oquab et al., 2023]. For each encoder, we considered the representation extracted (i) by the class encoder (class), (ii) by the average of all the other tokens (mean), or (iii) both concatenated (all). For each representation extracted we trained different heads, varying the number of hidden layers (1 or 2 layers with 256 nodes each with ReLU activation), learning rates (0.05, 0.005, 0.0005) for Adam optimizer [Kingma and Ba, 2014] (10 epochs) and target (independent double prediction of 'blue to focal' and 'orange to focal' grooming, or unique prediction of grooming either 'blue to focal' or 'orange to focal') via (binary) cross-entropy loss. We either discretized or not the output of the model, already in $[0, 1]$ due to the sigmoid final activation. For each configuration, we repeated the training with five different random seeds. A summary of the architectures and training description is in Appendix D.2.

**Evaluating**  For each trained model, we computed the binary cross-entropy loss, accuracy, balanced accuracy, and TEB on validation; and accuracy, balanced accuracy, TEB, and TEB using discretization on the full dataset $\mathcal{D} = \mathcal{D}_s \cup \mathcal{D}_u$ (where the average potential outcomes in the TEB are estimated by the sample mean). Since the ATE does not have a reference scale, for interpretation purposes, we replaced the TEB with Treatment Effect Relative Bias (TERB = TEB/ATE) in the visualizations.

### 5.2  CausalMNIST

CausalMNIST is a new synthetically generated visual dataset we designed for downstream treatment effect estimation. It is a colored manipulation of the MNIST dataset [LeCun, 1998], following an underlying generative process in agreement with the causal model assumed in our framework (see Figure 1). We explicitly controlled the ATE and generated 400 different samples from such a population (each one as large as the MNIST dataset, i.e., 60k images), allowing for Monte-Carlo simulations to accurately provide confidence intervals of our estimations. We omitted a comparison among pre-trained encoders since the visual task is relatively simple and can be solved directly by a simple convolutional neural network in a supervised fashion. A full description of the dataset is in Appendix E, together with its experiments, which align with our conclusions from ISTAnt.

## 6  Results

**Annotating criteria matter**  Theory suggests that biased annotating criteria (i.e., depending on the experimental settings) can lead to biased treatment effect estimation, wrongly retrieving the conditional treatment effect on unseen experimental settings. Figure 4 validates this observation, particularly in the few-shots regime. Despite the average estimation of the TEB is (almost) always biased, as illustrated in Table 1, the distribution for (per-video) random annotation is more centered towards 0. The benefits of random sampling are less obvious in the many-shots regime since $\mathcal{D}_u$ becomes less and less Out-of-Distribution. Still, this setting is rarely the case in practice since scientist hope to label $|\mathcal{D}_s| \ll |\mathcal{D}_u|$ frames to have a concrete advantage in their workflow.

**Encoder Bias**  Vanilla classification evaluation (e.g., accuracy, F1-score, etc.) well describes the goodness of a representation for a predictive downstream task. However, it is still unclear how to measure the goodness of a representation for a causal downstream task since we do not directly

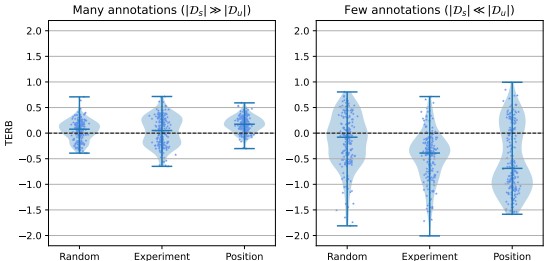

| Annot. | Criteria | $t$ | $p$-value |
|---|---|---|---|
| Many | Random | 3.581 | $4 \cdot 10^{-4}$ |
| | Experiment | 1.918 | 0.0564 |
| | Position | 14.982 | $\approx 0$ |
| Few | Random | -4.46 | $1.3 \cdot 10^{-5}$ |
| | Experiment | -13.417 | $\approx 0$ |
| | Position | -11.250 | $\approx 0$ |

Figure 4: Violin plots comparing the Treatment Effect Relative Bias (TERB) per annotation criteria in few and many annotations regime. Biased annotations lead to biased ATE estimation (i.e., TERB$\neq$0) and random annotation should be preferred.

Table 1: Two-sided $t$-test for $\mathcal{H}_0$ : $\mathbb{E}[\text{TERB}(f)] = 0$ over the 200 best models in overall Balanced Accuracy per splitting criteria. We found statistical evidence to reject the hypothesis that even the best models alone are unbiased for (almost) each annotation criterion.

| Encoder | Fréchet Distance (FD) | | |
|---|---|---|---|
| | Random | Experiment | Position |
| CLIP-ViT-L | $422.6 \pm 87.9$ | $461.2 \pm 151.7$ | $605.2 \pm 130.0$ |
| CLIP-ViT-S | $329.6 \pm 113.7$ | $341.7 \pm 120.2$ | $486.8 \pm 97.6$ |
| DINOv2 | $360.0 \pm 183.3$ | $413.0 \pm 222.5$ | $514.4 \pm 244.9$ |
| MAE | $275.0 \pm 20.9$ | $211.8 \pm 16.5$ | $\mathbf{760.9} \pm 122.4$ |
| ViT-L | $499.7 \pm 32.0$ | $503.4 \pm 108.6$ | $681.7 \pm 159.0$ |
| ViT-S | $308.7 \pm 69.8$ | $307.4 \pm 67.7$ | $423.9 \pm 103.9$ |

Figure 5: Scatter plot comparing the TERB and balanced accuracy in prediction among the 20 best models per 6 established encoders. Despite different downstream prediction performances, all the encoders (with excepts of MAE) lead to similar TERB (up to $\pm$ 50%).

Table 2: FD distance among $\mathcal{D}_s$ and $\mathcal{D}_u$, representing the average distribution distance ($\pm$ standard deviation) after normalization per encoder varying splitting criteria (e.g., few and many shots regime) and tokens considered. Representations with higher FD distance on position splitting (where the background changes the most) compared to the other splitting rely on more spurious correlations for our task (i.e., not minimal).

observe the ground truth (fundamental problem of Causal Inference). Even in our simple setting where we can easily identify the treatment effect over the whole population, it is not possible to condition just on a biased subsample (e.g., the validation set). Figure 5 shows clearly how the TERB doesn't correlate with balanced accuracy on the whole sample once it is sufficiently good (i.e., $> 0.9$). Even among models with balanced accuracy $> 0.95$ we estimated TERBs up to $\pm 50\%$, which can drastically lead to wrong causal conclusions. Among the different encoders, MAE is significantly underperforming all the others. We postulate the reason for this gap is that the masked reconstruction training leads to overly focus on background conditions instead of the comparatively small ants. Evidence for this hypothesis is reported in Table 2 where we observe that for 'position' splitting criteria, the Fréchet Distance between the extracted embeddings by MAE in $\mathcal{D}_s$ and $\mathcal{D}_u$ is maxima and significantly higher than for the other splittings, probably due to spurious correlation with the background which is indeed non changing as much for "random" and "experiment" splitting. Despite some (e.g., DINOv2 and CLIP-ViT-L) having better downstream predictive performances, the other encoders all have similar TERB ranges. New criteria to better estimate and bound the treatment effect bias already on validation and methodologies to unbias these models during training are required.

**Discretization Bias** We considered the absolute value of the TEB over all the $6\,480$ fine-tuned models, evaluating independently the models predicting both 'Blue to Focal' and 'Orange to Focal' grooming for a total of $9\,720$ evaluations. We tested ($t$-test):

$$\mathcal{H}_0 : \mathbb{E}[|\text{TEB}(f)|] = \mathbb{E}[|\text{TEB}(\mathbb{1}_{[0.5,1]}(f))|] \quad vs \quad \mathcal{H}_1 : \mathbb{E}[|\text{TEB}(f)|] < \mathbb{E}[|\text{TEB}(\mathbb{1}_{[0.5,1]}(f))|] \quad (5)$$

We found strong statistical evidence to confirm that discretizing the model outcome worsens treatment effect estimation ($t$ statistic=-10.42, $p$-value=$1.07 \cdot 10^{-25}$), confirming Theorem 3.1 .

**Prediction is not Causal Estimation**    Distinct statistical and causal objectives cannot be used as a proxy for one another. We already formalized this in Lemma 3.1 and partially observed it in Figure 5. In Figure 6, we systematically show it by comparing the rank-correlation among 1 620 models. We further observed that simply computing the TEB on a small validation is a better predictor of the TEB over the full dataset than the metrics focused on prediction accuracy (even on the full dataset). For the few-shot and experiment sampling (the most realistic), if we select the single best model on validation based on the TEB versus the accuracy, we underestimate the effect by 11% and 18%, respectively. While this is not perfect, is a significant improvement. We encourage future research to investigate theoretical generalization guarantees and new techniques to approximate the TEB on validation data.

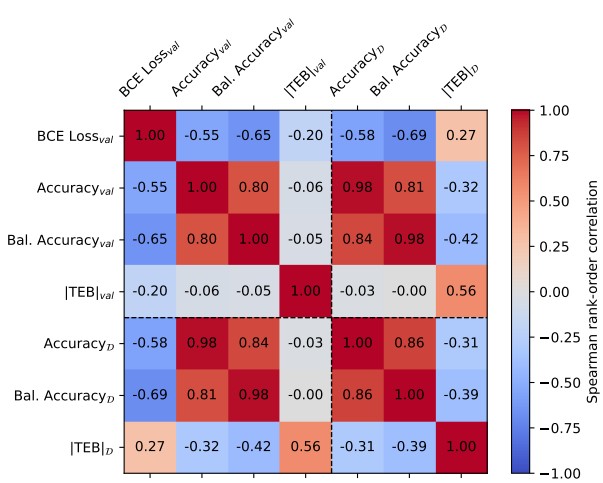

Figure 6: Spearman rank-order correlation matrix comparing different metrics for model selection on validation (subscript $val$) and over the full dataset (subscript $\mathcal{D}$). We considered all the 1 620 fine-tuned models to predict 'Blue to Focal' or 'Orange to Focal' grooming in few-annotations regime (i.e., $|\mathcal{D}_s| \ll |\mathcal{D}_s|$). Standard prediction metrics on validation correlate, but they are almost independent of the $|TEB|_{val}$. Similarly, they correlate with the prediction metrics on the full dataset but poorly predict the $|TEB|_{\mathcal{D}}$. On the other hand, $|TEB|_{val}$ is the most correlated metric with $|TEB|_{\mathcal{D}}$, unlike even the prediction metrics on the full dataset.

**Discussion**    Overall, our results clearly show that it is possible to leverage pre-trained deep learning models to accelerate the annotation of experimental data and obtain downstream causal estimates that are consistent with those from domain experts. At the same time, we find that experimental practices need to incorporate the specific needs of these causal downstream tasks. While our theoretical statements are "worst case scenarios" and only indicate that bias *can* arise (but does not always have to), we find empirical validation that it unfortunately easily manifests in practice. Remarkably, the fact that we performed and collected data within a randomized controlled trial, which is the best-case scenario of causal inference, did not alleviate the issue. Therefore, we can expect that the opportunities for bias can be even greater in observational settings, and even greater care is needed in model selection with the TEB and adaptation-time debiasing techniques.

## 7    Conclusions and Limitations

As AI models are increasingly used to answer scientific questions and support human decision-making, it is important to understand how design choices in machine learning pipelines affect the final results. In this paper, we took a closer look at the impact of pre-trained deep learning models in answering downstream causal treatment effect questions. We presented a real-world example in experimental behavioral ecology, creating the first-ever data set for treatment effect estimation from high-dimensional observations. Both theoretically and empirically, we observed that common choices, most notably discretizing the predictions and using in-distribution accuracy for model selection, can significantly affect the downstream conclusions. Two clear limitations of this work are that we did not do anything to the training to mitigate the bias, we kept the backbones frozen, and we did not incorporate the unlabelled data for semi-supervised training. Here, it would be very interesting to study how tools developed in the fairness literature can be extended to causal questions. For future benchmarks targeting scientific applications, we remark that it is vital to include the actual downstream question in the design of the data set. Otherwise, there is a risk that any model produced on that data may be unusable in practice, as it can bias the answer on an otherwise perfectly designed experiment. Finally, we would recommend that future work in causal representation learning starts from a clear downstream task like the one presented in this paper and works backward to reasonable assumptions. To facilitate this process, we will release our data set including all the experimental

variables, so that relevant future work on e.g., discovering confounding or semi-supervised effect discovery, can take place on a real problem.

**Acknowledgments**

We thank Piersilvio De Bartolomeis, and the full Causal Learning and Artificial Intelligence (CLAI) group at ISTA for the extremely helpful discussions. Riccardo Cadei was supported by a Google Research Scholar Award and a Google Initiated Gift to Francesco Locatello. We thank the Social Immunity team at ISTA, particularly Michaela Hönigsberger and Wilfrid Jean Louis, for supporting the ecological experiment and Farnaz Beikzadeh Abbasi, Luisa Fiebig and Martin Estermann for annotating ant behavior in ISTAnt.

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

# A  Proofs

## A.1  Proof of Lemma 3.1

**Lemma.** *Let $T \sim Be(p_T)$, $Y \sim Be(p_Y)$, $\boldsymbol{X} \sim \mathcal{P}^{\boldsymbol{X}}$ and let $f : \mathcal{X} \to [0,1]$ a model for $\mathbb{E}_Y[Y|\boldsymbol{X} = \boldsymbol{x}]$. Assuming (i) Ignorability (i.e., $T \perp Y|do(T = 1), Y|do(T = 0)$), (ii) $\mathbb{E}_{\boldsymbol{X}}[|\mathbb{1}_{[k,1]}(f(\boldsymbol{X})) = f(\boldsymbol{X})|] = 0$ where $k \in [0,1]$, and (iii) $f$ with accuracy $1 - \epsilon$, i.e., :*

$$\mathbb{P}\left(\mathbb{1}_{[k,1]}(f(\boldsymbol{X})) = Y\right) = 1 - \epsilon \quad \text{with } \epsilon \in [0,1], \qquad \text{(Classification Accuracy)}$$

*then $|TEB(f)| \leq \frac{\epsilon}{\min_t P(T=t)}$ and the worst-case $|TEB(f)| = \frac{\epsilon}{\min_t P(T=t)} \geq 2\epsilon$ is reached when all the misclassification over (or under) estimates the factual outcome in the smaller in size treatment group.*

*Proof.* Starting from the definition of Treatment Effect Bias and using assumption (i):

$$|\text{TEB}(f)| = |(\underbrace{\mathbb{E}_{\boldsymbol{X}|do(T=1)}[f(\boldsymbol{X})] - \mathbb{E}_{Y|do(T=1)}[Y]}_{\text{Interventional Bias under Treatment}}) - (\underbrace{\mathbb{E}_{\boldsymbol{X}|do(T=0)}[f(\boldsymbol{X})] - \mathbb{E}_{Y|do(T=0)}[Y]}_{\text{Interventional Bias under Control}})| =$$

$$= |(\underbrace{\mathbb{E}_{\boldsymbol{X}|T=1}[f(\boldsymbol{X})] - \mathbb{E}_{Y|T=1}[Y]}_{\epsilon_1}) - (\underbrace{\mathbb{E}_{\boldsymbol{X}|T=0}[f(\boldsymbol{X})] - \mathbb{E}_{Y|T=0}[Y]}_{\epsilon_0})| =$$

$$= |\epsilon_1 - \epsilon_0| \leq |\epsilon_1| + |\epsilon_0| \tag{6}$$

where

$$\epsilon_t := \mathbb{E}_{\boldsymbol{X}|T=1}[f(\boldsymbol{X})] - \mathbb{E}_{Y|T=1}[Y] \quad \forall t \in \{0,1\} \tag{7}$$

represent the overestimation of each conditional outcome expectation (i.e., conditional bias under treatment/control).

By assumption (iii) and the law of total probability:

$$\epsilon = \mathbb{P}\left(\mathbb{1}_{[k,1]}(f(\boldsymbol{X})) \neq Y\right) =$$
$$= \mathbb{P}\left(\mathbb{1}_{[k,1]}(f(\boldsymbol{X})) \neq Y|T=0\right) \cdot \mathbb{P}(T=0) + \mathbb{P}\left(\mathbb{1}_{[k,1]}(f(\boldsymbol{X})) \neq Y|T=1\right) \cdot \mathbb{P}(T=1). \tag{8}$$

By Jensen's inequality and linearity of the expected value:

$$|\epsilon_t| = |\mathbb{E}_{(\boldsymbol{X},Y)|T=t}[f(\boldsymbol{X}) - Y]| \leq$$
$$\leq \mathbb{E}_{(\boldsymbol{X},Y)|T=t}[|f(\boldsymbol{X}) - Y|] =$$
$$= \mathbb{E}_{(\boldsymbol{X},Y)|T=t}[|f(\boldsymbol{X}) - \mathbb{1}_{[k,1]}(f(\boldsymbol{X})) + \mathbb{1}_{[k,1]}(f(\boldsymbol{X})) - Y|] \leq$$
$$\leq \mathbb{E}_{(\boldsymbol{X},Y)|T=t}[|f(\boldsymbol{X}) - \mathbb{1}_{[k,1]}(f(\boldsymbol{X}))|] + \mathbb{E}_{(\boldsymbol{X},Y)|T=t}[|\mathbb{1}_{[k,1]}(f(\boldsymbol{X})) - Y|] =$$
$$= \mathbb{E}_{(\boldsymbol{X},Y)|T=t}[|f(\boldsymbol{X}) - \mathbb{1}_{[k,1]}(f(\boldsymbol{X}))|] + \mathbb{P}\left(\mathbb{1}_{[k,1]}(f(\boldsymbol{X})) \neq Y|T=t\right). \tag{9}$$

Combining Equation 8 and 9 we using the assumption (ii), for all $t \in \{0,1\}$ we have:

$$\epsilon \geq |\epsilon_0| \cdot \mathbb{P}(T=0) + |\epsilon_1| \cdot \mathbb{P}(T=1) - \mathbb{E}_{\boldsymbol{X}}[|\mathbb{1}_{[k,1]}(f(\boldsymbol{X})) = f(\boldsymbol{X})|] =$$
$$= |\epsilon_0| \cdot \mathbb{P}(T=0) + |\epsilon_1| \cdot \mathbb{P}(T=1) \tag{10}$$

And finally, combining this with Equation 6, we get:

$$|\text{TEB}(f)| \leq \frac{\epsilon}{\min_t P(T=t)}. \tag{11}$$

The bound we found corresponds to the worst-case scenario where we misclassify, only overestimating or only underestimating, always in the least probable treated group. Since $T$ is binary, then $(\min_t P(T=t)) > 0.5$, and so the thesis.

*Comment: Assumption (ii) is only used to find the worst-case scenario explicitly. Similar results can be stated bounding this discretization error.*

$\square$

## A.2 Proof of Theorem 3.1

**Theorem.** *Let $T \sim Be(p_T)$, $Y \sim Be(p_Y)$ and $\boldsymbol{X} \sim \mathcal{P}^{\boldsymbol{X}}$. For all $t \in \{0,1\}$, let $\hat{\tau}_n(\boldsymbol{X}, t)$ a succession converging in mean $L^1$ to $\tau(\boldsymbol{X}, t) := \mathbb{E}_Y[Y|\boldsymbol{X}, T = t]$, i.e.,*

$$\mathbb{E}_{\boldsymbol{X}}\left[|\hat{\tau}_n(\boldsymbol{X}, t) - \tau(\boldsymbol{X}, t)|\right] \xrightarrow{n} 0 \tag{12}$$

*Let $\hat{\tau}_n^*(\boldsymbol{X}, t) = \mathbb{1}_{[k,1]}(\hat{\tau}_n(\boldsymbol{X}, t))$ for all $n$ and $\tau^*(\boldsymbol{X}, t) = \mathbb{1}_{[k,1]}(\tau(\boldsymbol{X}, t))$, where $\mathbb{1}_{[k,1]} : \mathbb{R} \to \{0,1\}$ is the indicator function with threshold $k \in [0,1]$. Assuming $\tau(\boldsymbol{X}, t)$ having continuos CDF (i.e., $\mathcal{F}_{\tau(\boldsymbol{X},t)} \in \mathcal{C}^0$), then:*

$$\mathbb{E}_{\boldsymbol{X}}\left[|\hat{\tau}_n^*(\boldsymbol{X}, t) - \tau^*(\boldsymbol{X}, t)|\right] \xrightarrow{n} 0 \tag{13}$$

*but*

$$\mathbb{E}_{\boldsymbol{X}}[\tau^*(\boldsymbol{X}, t)] \neq \mathbb{E}_{\boldsymbol{X}}[\tau(\boldsymbol{X}, t)] \quad \forall k \in [0,1]/\bar{k}, \tag{14}$$

*i.e., they are generally different unless for a value $\bar{k} \in [0,1]$ depending on the distribution of $\tau(\boldsymbol{X}, t)$ (not observed in practice).*

*Proof.* Convergence in mean $L^1$ of that binarized estimator (Equation 13) follows directly from the fact the $L^1$ convergence implies convergence in distribution and Portmanteau Thereom (using the continuity assumption of $\tau(\boldsymbol{X}, t)$ CDF).

It only remains to show that the expectations of their limits generally differ. By developing the expected value of the $\tau^*(\boldsymbol{X}, t)$ we have:

$$\mathbb{E}_{\boldsymbol{X}}\left[\tau^*(\boldsymbol{X}, t)\right] = \int \mathbb{1}_{[k,1]}\tau(\boldsymbol{X}, t)\, dP_{\boldsymbol{X}} = \tag{15}$$

$$= \mathbb{P}(\tau(\boldsymbol{X}, t) \geq k) \neq \mathbb{E}_{\boldsymbol{X}}[\tau(\boldsymbol{X}, t)] \quad \forall t \in \{0,1\}, k \in [0,1]/\bar{k}. \tag{16}$$

where, by definition, $\bar{k}$ is the $\alpha$-quantile for $\tau(\boldsymbol{X}, t)$, with $\alpha = 1 - \mathbb{E}_{\boldsymbol{X}}[\tau(\boldsymbol{X}, t)]$ (uniqueness due to the continuity of $\tau(\boldsymbol{X}, t)$ CDF). □

# B   Additional Examples

## B.1   Full Description Example 1

Let's consider the following structural causal model in alignment with the generative process in Figure 1. Noises:

$$n_T \sim Be(p_T) \tag{17}$$

$$n_W, n_X \stackrel{\text{i.i.d.}}{\sim} \mathcal{N}(0,1) \tag{18}$$

$$n_Y \sim \mathcal{N}(0, \sigma_Y^2) \tag{19}$$

where $p_T \in (0,1)$ and $\sigma_Y^2 > 0$. and structural equations:

$$T := n_T \tag{20}$$

$$W := n_W \tag{21}$$

$$X := T + W + n_X \tag{22}$$

$$Y := \mathbb{1}_{[0,+\infty)}(X + n_Y) \tag{23}$$

By the Law of Total Probability and additivity of Gaussian distributions, it follows:

$$X \sim \mathcal{N}(p_T, 2 + p_T \cdot (1 - p_T)) \tag{24}$$

$$X|T = 1 \sim \mathcal{N}(1, 2) \tag{25}$$

$$X|T = 0 \sim \mathcal{N}(0, 2) \tag{26}$$

$$Y|T = 1 \sim Be\left(\phi\left(\frac{1}{\sqrt{2 + \sigma_Y^2}}\right)\right) \tag{27}$$

$$Y|T = 0 \sim Be(0.5) \tag{28}$$

$$Y^* := \begin{cases} 1 & \text{if } \mathbb{E}_Y[Y|X] > 0.5 \\ 0 & \text{if } otherwise \end{cases} \tag{29}$$

Then:

$$Y^*|T = 1 \sim Be(\phi(1/\sqrt{2})) \approx Be(0.76) \tag{30}$$
$$Y^*|T = 0 \sim Be(0.5) \tag{31}$$

And:

$$\text{AD}_{Y,T} = \phi\left(\frac{1}{\sqrt{2 + \sigma_Y^2}}\right) - 0.5 \tag{32}$$

$$\text{AD}_{Y^*,T} = \phi(1/\sqrt{2}) - 0.5 \neq \text{AD}_{Y,T} \tag{33}$$

Let $\hat{f}(x)$ a logistic regression estimator for $\mathbb{E}[Y|X = x]$ and:

$$\hat{f}^*(x) := \begin{cases} 1 & \text{if } \hat{f}(x) > 0.5 \\ 0 & \text{if } otherwise \end{cases} \qquad \forall x \in \mathbb{R}. \tag{34}$$

Setting $p_T = 0.5$ and $\sigma_Y^2 = 1$, we run a Monte-Carlo simulation with 50 different random seeds per sample size $n$, estimating the associational difference by the empirical associational difference (EAD), i.e., using the sample mean. The results are reported in Figure 3. We observe that $\hat{f}$ leads to a consistent estimate of the true associational difference, which corresponds to the ATE due to the Ignorability Assumption encoded in the causal model:

$$\text{EAD}_{\hat{f}(X),T} \xrightarrow{n} \text{AD}_{Y,T} = \text{ATE}_{Y,T} \tag{35}$$

and so:

$$\text{EAD}_{\hat{f}^*(X),T} \xrightarrow{n} \text{AD}_{Y^*,T} = \text{ATE}_{Y^*,T} \tag{36}$$

But, according to Theorem 3.1, its discretization is biased:

$$\text{AD}_{Y^*,T} - \text{AD}_{Y,T} = \left(\phi(1/\sqrt{2}) - \phi(1/\sqrt{3})\right) \approx 0.042 > 0 \tag{37}$$

and more generally, the stronger is the variance in the effect random noise $n_Y$, the bigger is the bias.

## C   ISTAnt

In our study, we analyzed grooming behavior in the ant *Lasius neglectus* in groups of three worker ants. The workers for the experiment were obtained from their laboratory stock colony, which had been collected from the field in 2022 in the Botanical Garden Jena, Germany. Ant collection and all experimental work were performed in compliance with international, national and institutional regulations and ethical guidelines. For the experiment, the body surface of one of the three ants was treated with a suspension of either of two microparticle types (diameter  5 μm) to induce grooming by the two nestmates, which were individually color-coded by application of a dot of blue or orange paint, respectively. The three ants were housed in small plastic containers (diameter 28mm, height 30mm) with moistened, plastered ground and the interior walls covered with PTFE (polytetrafluoroethane) to hamper climbing by the ants. Filming occurred in a temperature- and humidity-controlled room at 23°C within a custom-made filming box with controlled lighting and ventilation conditions. We set up nine ant groups at a time (always containing both treatments) and placed them randomly on positions 1-9 marked on the floor in a 3x3 grid with a distance of about 3mm from each other. Figure 7 illustrates the filming box and the displaying of the containers in each batch. The experiment was performed on two consecutive days. Videos were acquired using a USB camera (FLIR blackfly S BFS-U3-120S4C, Teledyne FLIR) with a high-performance lens (HP Series 25mm Focal Length, Edmund optics 86-572) in OBS studio 29.0.0 [Bailey, 2017] at a framerate of 30 FPS and a resolution of 2500x2500 pixels. From each original video (105x105 mm), we generated 9 individual videos

.mkv (each 32x32 mm, 770x770 pixels) by determining exact coordinates per container from one frame in GIMP 2.10.36 [Kimball and Mattis, 2023] and cropping of the videos with FFmpeg 6.1.1 [Tomar, 2006]. Annotation was performed over two consecutive days by three observers who had not been involved in the experimental setup or recording and were unaware of the treatment assignments to ensure bias-free behavioral annotation. They annotated the behavior of the ants during video observations, using custom-made software that saves the start and end frames of behaviors marked in a .csv file. In one of the videos, one of the nestmates' legs got inadvertently stuck to its body surface during the color-coding, interfering with its behavior, so the video was discarded. This left 44 videos from 5 independent setups (n=24 of treatment 1 and n=20 of treatment 2) of 10 minutes each for a total of 792 000 annotated frames. For each video, we provide the following information: the number of the set to which it belongs (1-5); the number of the position within the set reflecting the position of the ant group under the camera (1-9), for which we also provide 'coordinates' in the 3x3 grid (taking values -1/0/1 for both X and Y axis); treatment (1 or 2); the hour of the day when the recording was started (in 24h CEST); experimental day (A or B); the top left coordinate of the cropping square from the original video (CropX/CropY); the person annotating the video (given as A, B, C); the date of annotation (1: first day, 2: second day) and in which order the videos were annotated by each person (both reflecting a possible training effect of the person).

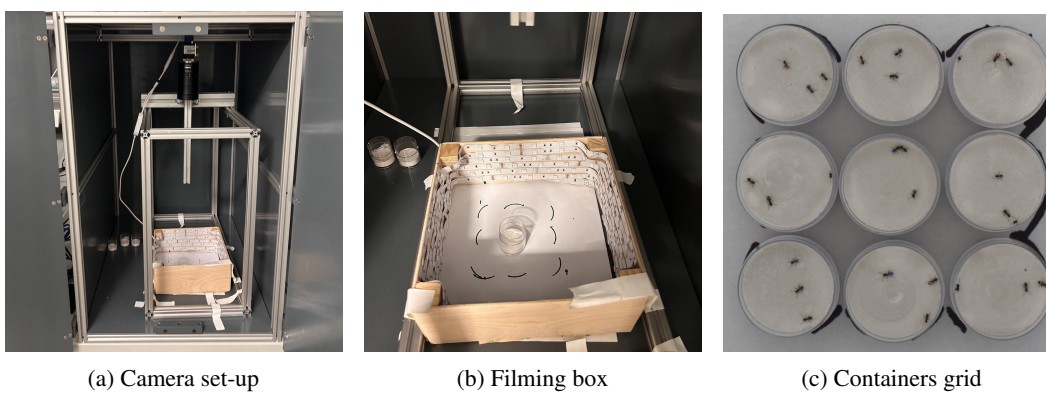

(a) Camera set-up                     (b) Filming box                     (c) Containers grid

Figure 7: Visualizations of the filming box set-up inside a temperature- and humidity-controlled room.

## D    Detailed Experimental Settings

In this section, we provide additional information on the experimental settings for the main experiments (on ISTAnt dataset). In particular, we describe the annotation splitting criteria selected, then the modeling choices and the training details. We run all the analyses using 48GB of RAM, 20 CPU cores, and a single node GPU (NVIDIA GeForce RTX2080Ti). The main bottleneck in the analysis is the feature extraction from the pre-trained Vision Transformers. We estimate 96 GPU hours to fully reproduce all the experiments described in the main paper.

### D.1    Splitting Criteria

Let $W_1 \in \{1, ..., 5\}$ representing the number of batch experiment and $W_2 \in \{1, ..., 9\}$ the relative position of a video inside its batch. We defined the annotation splitting criteria based on the value of the experiment settings $W_1$ and $W_2$, in agreement with Table 3.

where $\Omega = \{(1,2), (1,3), (2,4), (2,5), (3,1), (3,2), (4,3), (4,4), (5,9)\}$. For validation (used to generate the Figure 6) we consider 1 000 random frames from $\mathcal{D}_u$.

### D.2    Additional Models details

We extracted once the embedding of each frame in the dataset using a pre-trained encoder, and we fine-tuned multi-layer perceptron (MLP) heads for classification according to the training details reported in Table 4. We considered the following encoders for feature extraction, also report the corresponding Hugging Face code ID for reference:

| Annotations | Criteria | $\mathcal{D}_s$ | $n_s$ | $\mathcal{D}_u$ | $n_u$ |
|---|---|---|---|---|---|
| Many | Random | $(W_1, W_2) \notin \Omega$ | 42 000 | $(W_1, W_2) \in \Omega$ | 10 800 |
| | Experiment | $W_1 \neq 5$ | 42 000 | $W_1 = 5$ | 10 800 |
| | Position | $W_2 \neq 8$ | 46 800 | $W_2 = 8$ | 6 000 |
| Few | Random | $(W_1, W_2) \in \Omega$ | 10 800 | $(W_1, W_2) \notin \Omega$ | 42 000 |
| | Experiment | $W_1 = 1$ | 10 800 | $W_1 \neq 1$ | 42 000 |
| | Position | $W_2 = 1$ | 6 000 | $W_2 \neq 1$ | 46 800 |

Table 3: Annotation splitting criteria details for the extensive experiments on ISTAnt described in Section 5 and 6.

- ViT-B [Dosovitskiy et al., 2020]: `google/vit-base-patch16-224`
- ViT-L [Zhai et al., 2023]: `google/siglip-base-patch16-512`
- CLIP-ViT-B [Radford et al., 2021]: `openai/clip-vit-base-patch32`
- CLIP-ViT-L [Radford et al., 2021]: `openai/clip-vit-large-patch14-336`
- MAE [He et al., 2022],: `facebook/vit-mae-large`
- DINOv2 [Oquab et al., 2023]: `facebook/dinov2-base`

| Model/Hyper-parameters | Value(s) |
|---|---|
| Encoders | [CLIP-ViT-L, CLIP-ViT-S, DINOv2, MAE, ViT-L, ViT-S] |
| Encoder (token) | [class, mean, all] |
| MLP (head): hidden layers | [1,2] |
| MLP (head): hidden nodes | 256 |
| MLP (head): activation function | ReLU + Sigmoid output |
| Tasks | [all, or] |
| Dropout | No |
| Regularization | No |
| Loss | BCELoss (with positive weighting |
| Loss: Positive Weight | $\frac{\sum_{i=1}^{n_s} 1-Y_i}{\sum_{i=1}^{n_s} Y_i}$ |
| Learning Rates | [0.05, 0.005, 0.0005] |
| Optimizer | Adam ($\beta_1 = 0.9, \beta_2 = 0.9, \epsilon = 10^{-8}$) |
| Batch Size | 256 |
| Epochs | 10 |
| Seeds | [0,1,2,3,4] |

Table 4: Model and training details for the extensive experiments on ISTAnt described in Section 5 and 6.

Encoder (token) refers to which embedded tokens were considered for representation from each ViT. 'class' stands for the class taken, 'mean' for the mean of all the other tokens and 'all' for their concatenation. Task refers to which outcome we aimed to model directly: either the two independent grooming events ('Blue to Focal' and 'Orange to Focal') or the single grooming event ('Blue or Orange to Focal'). Overall, we finetuned:

$$n = n_{splitting\ criteria} \cdot n_{encoders} \cdot n_{tokens} \cdot n_{tasks} \cdot n_{hidden\ layers} \cdot n_{learning rates} \cdot n_{seeds}$$
$$= 6 \cdot 6 \cdot 3 \cdot 2 \cdot 2 \cdot 3 \cdot 5 = 6480 \tag{38}$$

heads.

# E  CausalMNIST

## E.1  Data generating process

To replicate the results on ISTAnt controlling for the causal model, we proposed CausalMNIST: a colored manipulated version of MNIST [LeCun, 1998], defining a simple causal downstream task (treatment effect estimation). Starting from MNIST dataset, we manipulated the background color $B$ of each image (1: green, 0: red), and the pen color $P$ (1: white, 0: black) to enforce the following Conditional Average Treatment Effect:

$$\mathbb{E}[Y|do(B=1), P=1] - \mathbb{E}[Y|do(B=0), P=1] = 0.4 \tag{39}$$
$$\mathbb{E}[Y|do(B=1), P=0] - \mathbb{E}[Y|do(B=0), P=0] = 0.2 \tag{40}$$

and Average Treatment Effect:

$$\mathbb{E}[Y|do(B=1)] - \mathbb{E}[Y|do(B=0)] = 0.3 \tag{41}$$

where $Y$ is a binary variable equal to 1 if the represented digit is strictly greater than $d \in \mathbb{R}$, 0 otherwise. Arjovsky et al. [2019] already proposed a colored variant of MNIST as a benchmark for robustness in a multi-environment setting, but without controlling for any causal model and presenting a causal downstream task. A simple interpretation of this new task is estimating the effect of the background on the chances of writing a big digit (i.e., greater than $d$).

To obtain a sample from such a population manipulating MNIST dataset, we converted each gray image into a RGB, coloring the background $B$ and the pen $P$ according to Bayes' rule:

$$P(B=b, P=p|Y=y) = \frac{P(Y=y|B=b, P=p) \cdot P(B=b, P=p)}{P(Y=y)} \quad \forall b, p, y \in \{0, 1\} \tag{42}$$

Since the digits in MNIST dataset are uniformly distributed:

$$Y \sim Be(p_Y) \tag{43}$$

where $p_Y = (9-d)/10$.

We then set:

$$B, P \overset{\text{i.i.d.}}{\sim} Be(0.5) \tag{44}$$

and:

$$P(Y=1|B=1, P=1) = p_Y + 0.2 \tag{45}$$
$$P(Y=1|B=0, P=1) = p_Y - 0.2 \tag{46}$$
$$P(Y=1|B=1, P=0) = p_Y + 0.1 \tag{47}$$
$$P(Y=1|B=0, P=0) = p_Y - 0.1 \tag{48}$$

in agreement with the Law of Total Probability and assuming $d \in \{1, 2, ..., 7\}$.

Overall, the final structural causal model can be summarized as follows:

- Noises (independent):

$$n_B \sim Be(0.5) \tag{49}$$
$$n_P \sim Be(0.5) \tag{50}$$
$$n_{\boldsymbol{X}} \sim P^{n_{\boldsymbol{X}}} \tag{51}$$
$$n_Y \sim P^{n_Y} \tag{52}$$

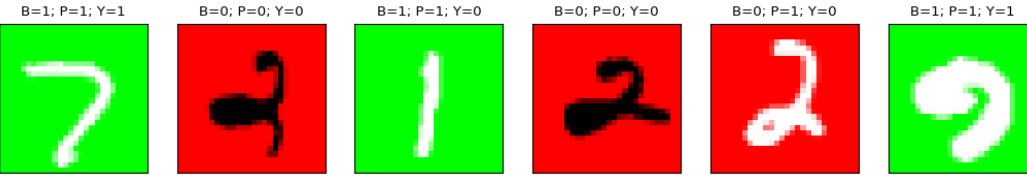

Figure 8: Random samples from CausalMNIST dataset.

- Structural equations:

$$B := n_B \tag{53}$$
$$P := n_P \tag{54}$$
$$\boldsymbol{X} := f_1(B, P, n_{\boldsymbol{X}}) \tag{55}$$
$$Y := f_2(\boldsymbol{X}, n_Y) \tag{56}$$

where $P^{n_{\boldsymbol{X}}}, P^{n_Y}, f_1$ and $f_2$ are unknown and characteristic of MNIST dataset. The corresponding causal model matches the setting described in Section 2 where $B$ represents the treatment $T$ and $P$ the experiment settings $\boldsymbol{W}$. In this analysis, we set $d = 3$. Six examples of colored handwritten digits from CausalMNIST are reported in Figure 8.

### E.2 Experimental Setting

**Annotation Sampling** Similarly to ISTAnt experiments, we compare the random annotation, where $S$ is assigned independently from $P$, and a biased annotation, where only the images with a black pen ($P = 0$) are annotated in both few and many annotation setting. The biased annotation criteria don't provide any information in $\mathcal{D}_s$ about the white pen ($P = 1$) CATE, and retrieving the annotations on $\mathcal{D}_u$ becomes mandatory. Unfortunately, a model could misclassify the new images under this covariate shift or hallucinate just for a specific treatment group (e.g., green background and white pen), leading to a biased estimate of the ATE. In Table 5, we summarize the 4 different annotation sampling proposed. For validation, we consider a random subsample of $\mathcal{D}_u$ as large as $\mathcal{D}_s$. Please observe that for the biased subsampling, not all the images with black pen (P=0) are allocated $\mathcal{D}_s$. Indeed, since $\mathbb{P}(P = 0) = 0.5 > \frac{n_s}{n_s + n_u}$ (in both few and many annotations regime), then $\mathcal{D}_u$ contains both images of hand-written digits in white and black.

| Annotations | Criteria | $\mathcal{D}_s$ | $n_s$ | $\mathcal{D}_u$ | $n_u$ |
|---|---|---|---|---|---|
| Many | Random | random | 12 000 | random | 48 000 |
| | Biased | only black (P=0) | 12 000 | the remaining (mixed) | 48 000 |
| Few | Random | random | 1 800 | random | 58 200 |
| | Biased | only black (P=0) | 1 800 | the remaining (mixed) | 58 200 |

Table 5: Annotation splitting criteria details for CausalMNIST experiments.

**Modeling** Since the vision task is relatively simple, i.e., extracting features from a pre-trained VisionTransformer is unnecessary, we don't replicate the comparison among different backbones, but we directly model the outcome through a simple Convolutional Neural Network. On the other hand, since we have control over the data-generating process, we generated CausalMNIST 100 times for each annotation sampling criteria using different random seeds, and we trained a Convolutional Neural Network (ConvNet) for each of them (i.e., Monte Carlo simulations). This way, comparing the different models, we still replicated the results for (i) data bias, (ii) discretization bias, and (iii) evaluation metrics already obtained for ISTAnt. The proposed ConvNet architecture consists of two convolutional layers followed by two fully connected layers. The first convolutional layer applies 20 filters of size 5x5 with ReLU activation, followed by a 2x2 max-pooling layer. The second convolutional layer applies 50 filters of size 5x5 with ReLU activation, followed by another 2x2 max-pooling layer. The output feature maps are flattened and passed to a fully connected layer with 500 neurons and ReLU activation. The final fully connected layer reduces the output to a single logit for binary classification (mapped to a probability through the sigmoid function). Table 6 reports a full description of the training details for such a ConvNet.

**Evaluation** We collected the same evaluation metrics for each training on both validation and the full dataset as described in Section 5.

| Hyper-parameters | Value(s) |
|---|---|
| Pre-Processing | Normalization |
| Dropout | No |
| Regularization | No |
| Loss | BCELoss |
| Loss: Positive Weight | No |
| Learning Rates | 0.001 |
| Optimizer | Adam $(\beta_1 = 0.9, \beta_2 = 0.9, \epsilon = 10^{-8})$ |
| Batch Size | 64 |
| Epochs | 6 |
| Seeds | $\{0,1, ..., 99\}$ |

Table 6: Training details for the ConvNets training on CausalMNIST.

### E.3 Results

We run all the analysis using $10$GB of RAM, $8$ CPU cores, and a single node GPU (`NVIDIA GeForce RTX2080Ti`). The main bottleneck of each experiment is re-generating a new version of CausalMNIST from MNIST dataset. We estimate a total of 6 GPU hours to reproduce all the experiments described in this section.

**Annotation criteria matter** Theory suggests that biased annotating criteria (i.e., depending on the experimental settings) can lead to biased treatment effect estimation, wrongly retrieving the conditional treatment effect on unseen experimental settings. Figure 9 validates this observation, and the results are validated via the $t$-tests reported in Table 7. Overall, the results perfectly align with the analogous discussion on ISTAnt.

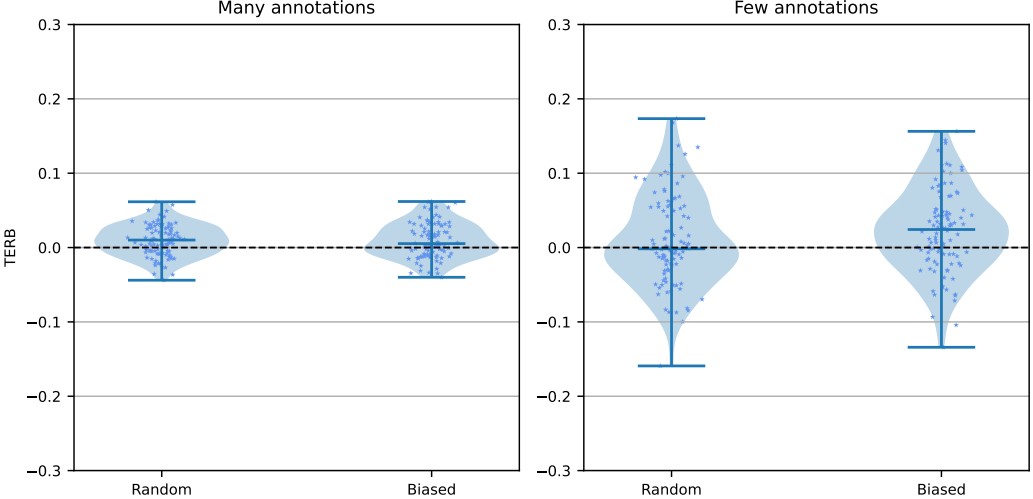

Figure 9: Violin plots comparing the Treatment Effect Relative Bias (TERB) per annotation criteria criteria in few and many annotations regime varying the seeds. Biased annotations lead to biased ATE estimation (i.e., TERB$\neq 0$) and random annotation should be preferred.

**Discretization bias** We considered the absolute value of the TEB over all the 400 experiments, and we tested ($t$-test):

$$\mathcal{H}_0 : \mathbb{E}[|\text{TEB}(f)|] = \mathbb{E}[|\text{TEB}(\mathbb{1}_{[0.5,1]}(f))|] \quad vs \quad \mathcal{H}_1 : \mathbb{E}[|\text{TEB}(f)|] \neq \mathbb{E}[|\text{TEB}(\mathbb{1}_{[0.5,1]}(f))|]$$
$$(57)$$

| Annot. | Criteria | $t$ | $p$-value |
|--------|----------|-----|-----------|
| Many | Random | 4.421 | $2.5 \cdot 10^{-5}$ |
|      | biased | 4.030 | $1.1 \cdot 10^{-4}$ |
| Few | Random | 1.607 | 0.111 |
|     | Biased | 3.911 | $1.7 \cdot 10^{-4}$ |

Table 7: Two-sided $t$-test for $\mathcal{H}_0 : \mathbb{E}[\text{TEB}(f)] = 0$. We found statistical evidence to reject the hypothesis that $f$ is unbiased for (almost) each annotation criterion.

We found no statistical evidence to reject the null hypothesis ($t$ statistic=1.188, $p$-value=0.235). Still, this result doesn't contradict Theorem 3.1, where we show that predictions, discretized and not, generally differ in expectation, but they can still be close (by chance). Some evidence of this undesired discretization effect can still be observed in the distribution of the $\text{TEB}(f)$ and $\text{TEB}(\mathbb{1}_{[0.5,1]}(f))$ as illustrated in Figure 10 for both random and biased sampling. In random sampling, in particular $\text{TEB}(\mathbb{1}_{[0.5,1]}(f))$ mean in random sampling is positive and 72.5% higher than $\text{TEB}(f)$ mean.

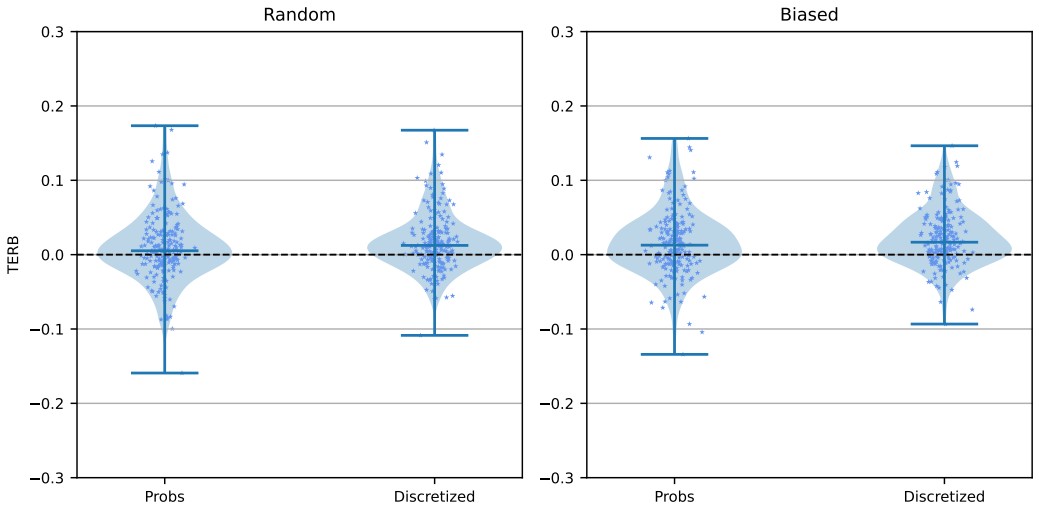

Figure 10: Violin plots of the TERB of the model (discretized or not) for both random and biased annotation sampling, varying number of annotations (few/many) and seeds.

**Prediction is not Causal Estimation** Distinct statistical and causal objectives cannot be used as a proxy for one another. We already formalized this in Lemma 3.1 and discussed it for ISTAnt dataset. In Figure 11 and 12, we systematically show it again for our new synthetic benchmark by comparing the rank-correlation among 200 ConvNets using random and biased sampling, respectively. Both matrices fully align with the discussion presented for the ISTAnt dataset in Section 6.

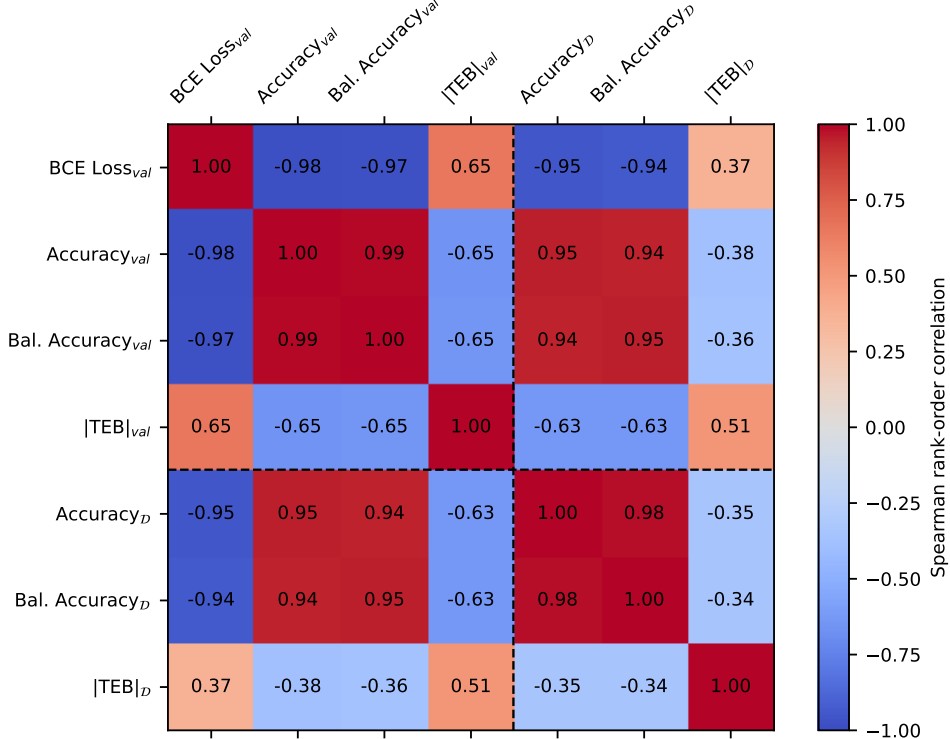

Figure 11: Spearman rank-order correlation matrix comparing different metrics for model selection on validation (subscript $val$) and over the full dataset (subscript $\mathcal{D}$). We considered all the 200 models trained with **random sampling**, varying the number of annotations (few and many) and seeds. Standard prediction metrics on validation strongly correlate, but they are less associated with $|\text{TEB}|_{val}$. Similarly, they correlate with the prediction metrics on the full dataset but poorly predict the $|\text{TEB}|_{\mathcal{D}}$. On the other hand, $|\text{TEB}|_{val}$ is the most correlated metric with $|\text{TEB}|_{\mathcal{D}}$, unlike even the prediction metrics on the full dataset.

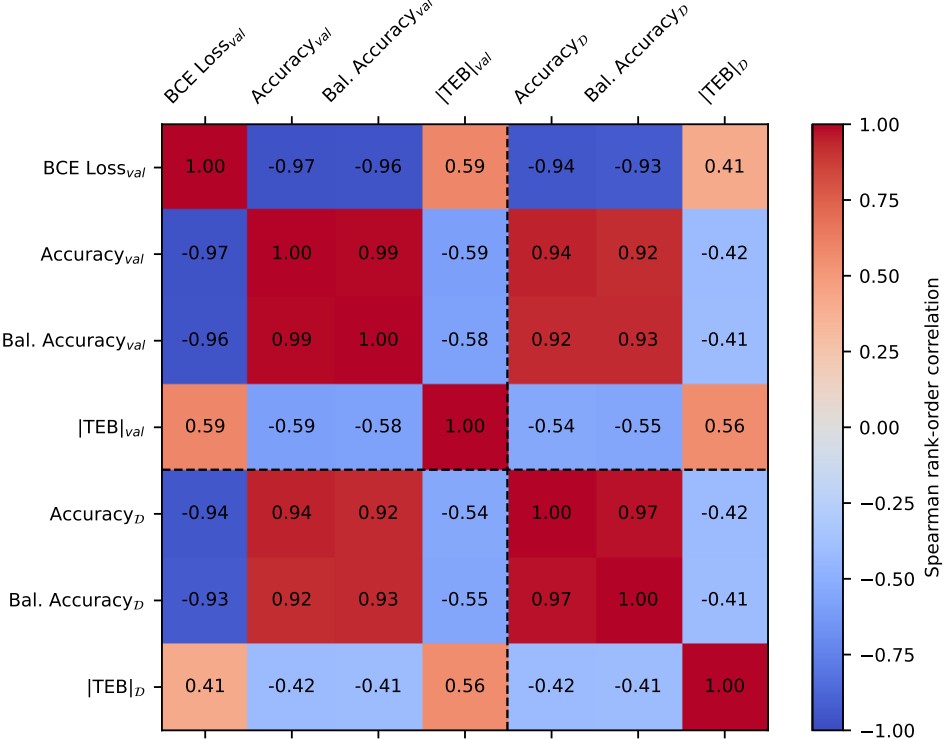

Figure 12: Spearman rank-order correlation matrix comparing different metrics for model selection on validation (subscript $val$) and over the full dataset (subscript $\mathcal{D}$). We considered all the 200 models trained with **biased sampling**, varying the number of annotations (few and many) and seeds. Standard prediction metrics on validation strongly correlate, but they are less associated with $|\text{TEB}|_{val}$. Similarly, they correlate with the prediction metrics on the full dataset but poorly predict the $|\text{TEB}|_{\mathcal{D}}$. On the other hand, $|\text{TEB}|_{val}$ is the most correlated metric with $|\text{TEB}|_{\mathcal{D}}$, unlike even the prediction metrics on the full dataset.

