# OpenReview forum: "Smoke and Mirrors in Causal Downstream Tasks"
_NeurIPS.cc/2024/Conference — NeurIPS 2024 poster_

### Official Review · Reviewer_Zfn9 · 2024-06-20

**Soundness:** 3
**Presentation:** 3
**Contribution:** 3
**Rating:** 6
**Confidence:** 3

**Summary:**

the paper makes a deep dive into the various types of biases that can arise in RCTs that would invalidate causal estimants

**Strengths:**

- Interesting paper that would greatly benefit the discussions of the community
- Identifies crucial sources of bias, offers interesting potential solutions to them
- Experimentally shows the biases in a large number of models

**Weaknesses:**

- The literature review only focuses on the last 2 years and ignores prior works that are still relevant
- there is little theoretical argumentation in the proposed solutions
- there is little experimental evidence on ways to overcome the identified problems

**Questions:**

- not much to be honest, the identification and codification of the RCT biases and how these effect causal estimates makes the paper pass the bar for me

**Limitations:**

- adequately addressed

---

> ### Author Rebuttal · Authors · 2024-08-05
>
> **Weakness 1**: *Only reviewing the last 2 years' literature*
>
> We thank the reviewer for the feedback. We agree that we could improve on the literature review, especially with respect to classical causal inference works. We will add them to the related works. We would appreciate it if the reviewer had pointers in mind we should not miss.
>
> **Weakness 2 and 3**: *little theoretical argumentation/experimental evidence for solutions*
>
> We agree that the current paper largely focuses on showcasing the challenges. While we offer principled solutions in some aspects, e.g., discretization bias, we lack in others, e.g., how to do model selection, how to de-bias, etc. Given the breadth of the paper (also mentioned by Reviewer Erid as a limitation: “*The biggest weakness of this paper is that it tries to do a bit too much*”), we think it’s fair to leave solutions for future work and focus the narrative on clearly conveying what the challenges are. We hope this will attract future contributions by the broader community, bridging causal inference, representation learning for ecology, and causal representation learning.

---

> > ### Comment · Reviewer_Zfn9 · 2024-08-12
> > **Acknowledgement**
> >
> > I acknowledge that I have read the authors rebuttal, and I maintain my score of acceptance

---

### Official Review · Reviewer_WAFK · 2024-07-10

**Soundness:** 3
**Presentation:** 3
**Contribution:** 3
**Rating:** 7
**Confidence:** 1

**Summary:**

This paper theoretically reveals that many common choices in the literature may lead to biased estimates. To test the practical implications of these considerations, this paper recorded the first real-world benchmark for causal inference downstream tasks on high-dimensional observations through an RCT studying how garden ants (Lasius neglectus) respond to microparticles applied to their colony members by hygienic grooming. By comparing 6,480 models fine-tuned from state-of-the-art visual backbones, they found that the sampling and modeling choices significantly affect the accuracy of the causal estimates, and that classification accuracy is not a reliable proxy for this accuracy.

**Strengths:**

1. To facilitate future research on representation learning for causal downstream tasks, the authors have formulated representation desiderata to obtain accurate estimates for causal queries, along with best practices.
2. The authors have designed and collected a new dataset, conducted extensive experiments, and proposed a new synthetic benchmark.
3. This paper is well-structured, with thorough theoretical derivations and experimental discussions.

**Weaknesses:**

1. The link to the data set provided by the author cannot be accessed.
2. The author's presentation of the provided data set is a little bit limited, especially in terms of visualizations. It would be helpful to include some actual images of the physical data.

**Questions:**

1. Could authors provide more details on the dataset collection process and explain why ants were used as an example?
2. Could the method proposed in this paper be generalized to other fields, beyond the ecological example mentioned?

**Limitations:**

The authors thoroughly discussed the limitations of this paper.

---

> ### Author Rebuttal · Authors · 2024-08-05
>
> **Weakness 1**: *broken link*
>
> We are sorry for the inconvenience, the anonymous hosting account was eventually disabled without further notice due to inactivity. We added a new link to review the dataset in our general answer.
>
> **Weakness 2**: *limited description and visualization of the dataset*
>
> Figure 2 offers an example of two random scenes with different behavior. These are actual images from the data set. Refer to the new link to investigate it directly. For more details on the dataset description and generating process, consult Appendix C, to which we will add more pictures of the data set and the recording setup in the final version of the paper.
>
> **Questions 1 and 2**: *do you generalize beyond ecology? Why Ants? More details on data collection?*
>
> * Yes, we generalize beyond ecology. Our discussion and conclusions are general, extending to **any high-dimensional causal inference problem where the outcome of interest is observed only through a low-level signal (i.e., pixels)**, expensive or challenging to manually annotate. To make this case, we also repeated the analysis on a synthetic data set we proposed (Causal MNIST) in the Appendix.
>
> * To showcase the challenges involved in treatment effect estimation from high-dimensional data, we performed a real-world randomized controlled trial. We chose ecology and, in particular, Ants because (1) they are small enough to experimentally study their collective behavior on a lab bench and (2) they exhibit complex interactions that are modulated by their environment. This is a big field of study in itself, which would benefit from AI methods to increase the scale at which experiments can be analyzed, but it has not bridged to the AI community yet. At the same time, our analysis is more general, as explained above. This will be clarified in the final version of the paper.
>
> * As discussed in "weakness 2" above, we will add more details on the data collection in Appendix C, together with pictures of the data recording setup and more samples from the data set.

---

> > ### Comment · Reviewer_WAFK · 2024-08-10
> >
> > Thanks for the author's rebuttal. I will keep my score.

---

### Official Review · Reviewer_Erid · 2024-07-11

**Soundness:** 3
**Presentation:** 3
**Contribution:** 4
**Rating:** 8
**Confidence:** 3

**Summary:**

This paper considers the task of causal effect estimation $P(Y|do(T))$, where the treatment is mediated through a (potentially high-dimensional) observation $X$.
Additionally, a semi-supervised setting is studied, where labels $Y$ are only available in a subset of the data.
A set of possible biases affecting the treatment effect estimation that can arise in this setting are discussed together with possible mitigation strategies.
Two datasets are introduced as new benchmarks:
(i) An Ant Dataset, where groups of ants are filmed under the treatment of putting microparticles on the body surface. Grooming behaviour is observed through video recordings.
(ii) CausalMNIST: A dataset that adds colour coding to MNIST data and adds a downstream causal task.
The biases discussed above are empirically evaluated on a large battery of models that are fine-tuned in few- and many-shot learning.

**Strengths:**

I think this paper has many great contributions and approaches the questions it tries to answer systematically.

- The biggest contribution IMO is the ant video dataset. Causal representation suffers from overly simplistic settings and restrictive assumptions. Such datasets have the potential to move this field closer to practically useful applications and out of the ivory tower of identifiability under very restrictive assumptions.
- I love Sec. 3 and the fact that it is written as a sort of self-help book on causal effect estimation. This was refreshing to read.
- The writing is superb.
- The systematic evaluation of a large number of different models is great.

**Weaknesses:**

The biggest weakness of this paper is that it tries to do a bit too much.
Given the NeurIPS page limit, this leads to things being condensed to a point that makes it difficult to follow at times.
For example, Theorem 3.1 tries to convey an idea without properly defining most of its elements, and IMHO it fails to convey much at all.
Without looking up the formal definition in the appendix, it's difficult to understand.

**Questions:**

- Sec. 5.1: What exactly are $X$ and $Y$ for this dataset?

**Limitations:**

- "Model bias from the encoder choice" (L146): This point is a bit wishy-washy and, as it is, it has limited use to the practitioner. It is too generic and doesn't offer much practical guidance on how to e.g. "attempt to quantify its biases". Maybe this space would be better spent on expanding on other parts of the paper.

---

> ### Author Rebuttal · Authors · 2024-08-05
>
> **Weakness**: *doing too much (e.g., informal Thm. 3.1 difficult to follow)*
>
> Thank you for pointing this out. Making Section 3 concise and accessible to different communities was indeed a not trivial task. We agree that Thm 3.1 is not as crisp as it could be. We suggest making the definition of Thm. 3.1 more technically precise in the main paper, followed by a high-level intuition, while still keeping the complete statement in the appendix.
>
> **Question**: *X and Y definition*
>
> Per frame, X is what happens in the real world and is then measured with the camera view (RGB image) of the experiment, and Y is the current behavior (e.g., a blue ant grooming the focal). Figure 2 illustrates these but we will further clarify it in the main text.
>
> **Limitation**: *Paragraph ‘Model bias from the encoder choice’ a bit wishy-washy*
>
> Thank you for pointing this out. Depending on space constraints, we suggest either integrating a more technical discussion or making space for other content (e.g., on the discretization bias).

---

> > ### Comment · Reviewer_Erid · 2024-08-08
> >
> > Thank you for the clarifications.
> >
> > On the limitation: I'm happy with either solution (if you expect me to weigh in on which option to choose).

---

> > > ### Author Response · Authors · 2024-08-08
> > >
> > > Thank you for the feedback. We are also happy either way and will decide based on the amount of space we have available as we prepare the next revision.

---

### Official Review · Reviewer_ZnDR · 2024-07-14

**Soundness:** 2
**Presentation:** 3
**Contribution:** 2
**Rating:** 6
**Confidence:** 2

**Summary:**

The paper explores the challenges associated with using machine learning, particularly deep learning, to estimate causal treatment effects from high-dimensional data, such as images, in Randomized Controlled Trials (RCTs). The authors point out that standard practices in machine learning, such as selecting models based on accuracy and discretizing predictions, can lead to biases in causal estimates. To substantiate their claims, they provide both theoretical analyses and empirical results. They introduce a novel real-world benchmark dataset that involves the grooming behavior of ants, which they assert is the first benchmark designed for causal inference downstream tasks. The authors stress the necessity of incorporating causal downstream tasks into benchmark design and offer guidelines to enhance causal inference in scientific applications through machine learning.

**Strengths:**

1. The paper addresses an important and under-explored area of causal inference, which will facilitate further research.

2. The authors provide some theoretical analysis to identify potential sources of bias in treatment effect estimation using deep learning models.

3. The paper is well-written and well-organized.

4. The experiments involve fine-tuning thousands of machine learning models with various encoders, enhancing the reliability of the conclusions.

**Weaknesses:**

I am not very familiar with this topic, so I may adjust my rating based on further discussions with the authors and reviewers.

1. The main theoretical analysis focuses on binary classification, which may not generalize to many real-world settings. On line 251, the authors claim that Theorem 1 is validated on their dataset. Could you provide more details on how Theorem 1 is validated? I found it difficult to follow.

2. The setting of this paper differs from many causal representation learning algorithms, making it challenging to directly evaluate those algorithms on this benchmark. This may limit the applicability of the proposed benchmarks. Could you apply some existing causal representation learning methods to the proposed datasets and compare the conclusions? The experiments only test different pre-trained models, but it would be more compelling to demonstrate that this new benchmark can help answer causal questions with training in a causal manner.

3. The authors frequently mention causal inference downstream tasks. Can you further elaborate on why the proposed dataset is superior to other benchmarks for answering causal questions? The experiments primarily show that bias can arise, but I am not fully convinced that the proposed dataset is better than the existing ones.

4. While the authors acknowledge limitations such as keeping the backbones of models frozen and not incorporating semi-supervised learning, these constraints might affect the generalizability of their findings.

**Questions:**

Please see the weaknesses below. I may reconsider my score after further discussion with the authors and reviewers. My main concern is the significance of the proposed benchmark and its utility in evaluating different causal representation algorithms. Why is this dataset better suited for answering causal questions and reducing bias compared to existing datasets?

**Limitations:**

Yes.

---

> ### Author Rebuttal · Authors · 2024-08-05
>
> **Weakness 1**: *Why only binary outcome*
>
> Thank you for the constructive feedback. We realize this point was overall not sufficiently remarked in the draft and we will stress it in the conclusion section in the camera-ready version. The two key points are:
> * Our discussion refers to the binary outcome case, in **agreement with** the majority of the causal inference **literature** [Robins et al., 2000, Samet et al., 2000], assuming an effect can either manifest or not. All the more so, this **simplification further values our thesis** that *factual estimation via representation learning for causal inference is subtle even in the simplest possible setting* (e.g., randomized controlled trial with binary outcome)
>
> * Excluding the Discretization Bias,  **all the other conclusions naturally extend to the continuous setting** (the potential outcomes are still well-defined).
>
>
> **Weakness 1bis (typo?)**: *Validation of Theorem 1 at line 251*
>
> We are unsure which theorem the reviewer refers to. We do not have theorem one, and no theorem is mentioned on line 251. Perhaps there is a typo or mistake in the question; otherwise, could you please clarify the pointers?
>
> **Weakness 2**: *Why not benchmark using CRL methods*
>
> Their assumptions are far from our real-world setting. For two main reasons, we cannot change the data's assumptions to fit some existing methods.
>
> 1) **Our benchmark addresses a real-world problem and workflow in experimental ecology**: our data comes straight out of an actual randomized controlled trial. Changing the assumptions generally means changing the design of the trial, which may invalidate the scientific conclusions one can draw from the results.
>
> 2) It is **technically infeasible** to incorporate the usual assumptions from CRL in a meaningful way. For example, interventional methods assume the possibility of performing interventions; in our case, they should be on the effect. This is the collective behavior of living ants, which we cannot directly intervene on. Multi-view settings would require a second modality, which we do not have.
>
> Overall, we hope that this paper will **encourage the causal representation learning field to start from a real problem and work out solutions, involving specifying what assumptions are reasonable for the task** and not the other way around. Building completely new causal representation learning methods for this application is something we highly encourage for future work, but is beyond the scope of the current paper.
>
> **Weakness 3 and Question 1**: *Novelty of the data set*
>
> Our data set and accompanying analysis are novel across three different lines of work:
>
> * In *treatment effect estimation*, effects are always assumed to be directly measured [Rubin, 1978], but this is not the case in scientific experiments. How to leverage machine learning for this is an open question for which this data set is the first of a kind.
>
> * In *representation learning*, especially for ecology, existing benchmarks focus on estimating accuracy, e.g., [Sun et al. 2023]. We have thoroughly demonstrated that even highly accurate models cannot be blindly used to draw causal conclusions.
>
> * In *causal representation learning*, data sets are built to match the assumptions of specific algorithms. This practice has arguably led to a scarcity of applications, as also highlighted in this paper. Instead, we start from an exemplary real-world causal downstream task and accessibly explain what properties are needed for new methods to be successful in practice.
>
> **Weakness 4**: *frozen backbone and no semi-supervised*
>
> We generally agree on the importance of repeating our analysis for different approaches and proposing new solutions, but the main objective of this paper is to crisply formulate this new problem and raise awareness of its subtle challenges. We remark that Reviewer Erid already pointed out that *this paper “tries to do a bit too much”*.
> Addressing possible solutions is beyond the scope of this paper, and we explicitly call for new methods in the conclusions. Our hope is that this paper will serve as an introduction for researchers in the three fields of causal inference, representation learning for ecology, and causal representation learning, to the challenge of applying representation learning for causal downstream tasks.

---

> > ### Comment · Reviewer_ZnDR · 2024-08-12
> >
> > Sorry for the late reply. Thank you so much for the detailed rebuttal. Most of my concerns are addressed. Regarding my first question, I meant Theorem 3.1—sorry for the typo. In lines 301-302, I saw that Theorem 3.1 is empirically validated, but adding more analysis could be beneficial. After reading other reviews, I cannot find significant issues, and I am willing to increase my score to 6 while keeping the confidence level at 2.

---

### Author Rebuttal · Authors · 2024-08-05

We thank the reviewers for their consideration of our paper and for their feedback. The consensus appears to be that the writing is “*superb*” [Erid] and “*well-structured*” [ZnDR, WAFK], “*addressing an important and under-explored area of causal inference*” [ZnDR] bringing “*many great contributions*” [Erid] “*which will facilitate future research*” [ZnDR, WAFK] and “*benefit the discussions of the community*” [Zfn9].

We will address their individual questions and comments separately.

**Link to the data set:**
We apologize if the link to the dataset eventually broke due to inactivity on the preliminary and anonymous drive account we created to share the data with the reviewers. We reshare here a new anonymous Figshare link: https://figshare.com/s/0970e149cfe72089c771.

---

### Decision · Program_Chairs · 2024-09-25

**Decision:**

Accept (poster)

**Comment:**

This paper investigates the bias when estimating the causal quantities in high dimensional RCTs. The author identifies the bias led by the common design choice. More importantly, the author provide a high-dimensional benchmark dataset for causal inference problem, which is a valid contribution to the causal community.